# Individual Preferences for COVID-19 Vaccination under the China’s 2021 National Vaccination Policy: A Discrete Choice Experiment Study

**DOI:** 10.3390/vaccines10040543

**Published:** 2022-03-31

**Authors:** Siyuan Wang, Stephen Nicholas, Elizabeth Maitland, Anli Leng

**Affiliations:** 1Faculty of Business and Economics, University of Melbourne, Melbourne, VIC 3051, Australia; siyuanw2@student.unimelb.edu.au; 2Australian National Institute of Management and Commerce, Sydney, NSW 2015, Australia; stephen.nicholas@newcastle.edu.au; 3Newcastle Business School, University of Newcastle, Newcastle, NSW 2308, Australia; 4School of Management, University of Liverpool, Liverpool L697ZH, UK; e.maitland@liverpool.ac.uk; 5School of Political Science and Public Administration, Shandong University, Qingdao 266237, China; 6Center for Health Preferences Research, Shandong University, Jinan 250012, China

**Keywords:** vaccination preference, COVID-19, post-2021 national vaccination policy, vaccination preference changes

## Abstract

(1) Background: Since China’s national vaccination policy announcement in January 2021, individual vaccination preferences related to vaccine characteristics, social relationships, sociodemographic characteristics and cognition remain opaque. This study aims to investigate vaccination preferences regarding these attributes, and to assess changes in individual vaccine preferences since the pre-2021 emergency vaccination phase. (2) Methods: The two-part questionnaire surveyed 849 individuals between May and June 2021 in Qingdao, China. The survey contained eight binary choice tasks that investigated preference trade-offs. Respondents’ sociodemographic characteristics, including age, sex, urban/rural residence, income, education and whether living with the young or old, were also collected. Conditional logit, mixed logit and latent class models were used to quantify preference utility and identify preference heterogeneity. (3) Results: Vaccine effectiveness, vaccine side effects, duration of protection and probability of infection all significantly affected vaccination utility. Preference heterogeneity based on individual social relationships and sociodemographic characteristics were also established. Marginal analysis showed that compared to the pre-2021 phase, individuals’ preferences had shifted towards vaccines with longer protection periods and better accessibility. (4) Conclusion: This study will inform the full rollout of China’s 2021 national vaccination program and provide valuable information for future vaccination policy design to meet resurgent COVID-19 risks.

## 1. Introduction

Between January 2020 and December 2021, there have been 2.8 billion confirmed global cases of COVID-19, with 5.4 million recorded deaths [1]. Since the initial Alpha outbreak, mutated variants (Delta and Omicron) have become the dominant strands of the virus and pose a growing global public health challenge [2,3]. Vaccination continues to be the most effective way to combat COVID-19 [4,5,6,7], with WHO reporting 8.6 billion doses of COVID-19 vaccines administered as of December 2021 [1]. Many countries have introduced booster doses to combat COVID-19 variants and prolong vaccine effectiveness [8,9]. For China, vaccination is the central focus for COVID-19 prevention. While China had administered 2.7 billion vaccine doses by December 2021 [10], further vaccinations are required for disease control, as COVID-19 is resurgent in China and abroad [11,12]. The Chinese government set a two-dose vaccination target for 80% of the population by the end of 2021 [13], with booster doses implemented in many provinces to contain resurgent COVID-19 outbreaks [14].

Understanding individual vaccination preferences is vital to facilitate vaccine roll-outs and increase vaccine coverage. Previous vaccination studies have identified vaccine efficacy, safety, history and costs, social relationships, cognition (risk–benefit vaccine perceptions), trust in healthcare systems, openness to experience and sociodemographic characteristics as important factors influencing individual vaccine uptake [15,16,17,18,19,20,21,22,23,24,25]. Prior to the approval of the first Sinopharm vaccine in 2021 [26], Chinese public opinion surveys of vaccine acceptance rates ranged between 36.4% and 94.3% [21,27,28]. Studies found that vaccine hesitancy caused by misinformation, pre-existing medical conditions, mistrust towards medical institutions and governments, concerns for vaccine efficacy, fear of vaccine adverse effects and respondents’ sociodemographic characteristics were the main reasons behind vaccination reluctance [29,30,31,32,33]. Other studies identified healthcare workers, nurses, college students and parents with children under 18 as specific groups requiring targeted pro-vaccine communication to address vaccine hesitancy [16,31,34,35,36,37,38,39,40,41,42,43,44,45,46]. Health belief models related Chinese vaccine hesitancy to individual cognition factors, such as perceived benefits, perceived risks and perceived barriers [33,37]. During the pre-2020 study period, it was unclear whether China’s vaccinations would be free [38], and research has found that costs significantly affect individual willingness to vaccinate [15,22,38,39].Studies investigating pre-2021 individual vaccination preference trade-offs found that vaccine efficacy, vaccine-related side effects, vaccination sites, duration of protection, access to vaccine, number of doses required and percentages of acquaintances vaccinated impacted individual vaccination preferences [40,41].

Since the above studies, China approved the Sinopharm vaccine in December 2020 and the Sinovac vaccine in February 2021, which account for the vaccines with the majority of the market share for general use [26,38,42,43]. Second, on 9 January 2021, the National Health Commission of China (NHS) decreed a national free and voluntary COVID-19 vaccination policy [44]. Under the 2021 national vaccination policy, free vaccination increased individuals’ willingness to uptake [45,46]. However, various studies have reported that vaccine hesitancy caused by misinformation, lack of trust, concerns regarding vaccine efficacy and safety, cognition and sociodemographic characteristics continued to deter vaccine uptake [47,48,49,50,51,52,53,54]. Under the 2021 national vaccination policy, discrete choice experiment (DCE) studies assessed individual vaccination preferences based on vaccine attributes, such as effectiveness, adverse effects, protection period and cost [55,56,57,58]. One constraint was that these studies were limited by the absence of variables on individuals’ social relationships and cognition in the vaccination decision. Additionally, there is a lack of knowledge on how individual vaccination attitudes might have changed between the pre-2021 and post-2021 national policy periods.

To address these lacunae, we undertook a DCE to assess individual vaccination preferences under China’s 2021 free national vaccination policy. Our study is one of the first to assess vaccination preferences under China’s national vaccination policy, considering not only vaccine characteristics, but social relationships and cognition, such as perception of risk. Social relations are an individual’s immediate relationships, defined in this study as the percentage of acquaintances vaccinated and living with children and the elderly. Our study also addresses how public acceptance of the COVID-19 vaccine differed across different phases of the epidemic marked by China’s pre- and post-2021 national vaccination policy.

Our study location was Qingdao, a high per capita income and major transport and manufacturing port city in eastern Shandong Province. Administering 217,185 dosages pre-2021 [59,60,61,62,63], and 24.1 million vaccine dosages by January 2022 under China’s 2021 national vaccination policy [64], Qingdao and Shandong also faced local COVID-19 outbreaks [65]. Understanding vaccination preferences in China is particularly important given the risk of the Omicron variant, which was resurgent in Hong Kong in February 2022, with, under China’s zero-COVID-19-tolerance policy, flare-ups contained in mainland China. Our study will inform the rollout of China’s national vaccination program and provide valuable information for future vaccination policy design to meet recurring COVID-19 risks.

## 2. Materials and Methods

### 2.1. Identification of Attributes and Levels

Based on the existing literature [41,66,67,68,69], expert interviews and a pilot study, six key attributes (vaccine effectiveness, vaccine-related side effects, vaccination sites, duration of vaccine protection, acquaintances vaccinated and perception of risk) were identified [41]. The probability of oneself and close acquaintances (friends and family) being infected with COVID-19 was assigned 100/100,000, 6/100,000 and 1/100,000 probabilities. Other attribute levels followed the COVID-19, seasonal influenza, H1N1 and hepatitis B vaccine literature [41,66,67,68,69]. Vaccine effectiveness comprised three levels: 40%, 60% and 85%. Side effects consisted of three levels: 1/100,000, 10/100,000 and 50/100,000. Reflecting the nation’s healthcare provider system, vaccination sites were distinguished by level 1 (village clinic or community health station), level 2 (township or community health centre) and level 3 (county hospital or above) facilities [41]. The duration of protection was 6 months, 1 year and more than 2 years, and the percentage of acquaintances vaccinated was set at 3 levels: 30%, 60% and 90% [41]. A list of attributes and levels are shown in Table 1.

### 2.2. Experimental Design

Consistent with Leng et al. [41], D-efficient partial profile design was used to guarantee that preference parameters can be estimated with maximal precision. Twenty-four hypothetical two-alternative choice tasks were created. For each choice task, respondents were asked to choose which vaccine they would prefer. To reduce the cognitive burden on respondents, these 24-choice tasks were divided randomly into 3 different versions of the questionnaire. Each version contained 8 choice tasks. An example of a choice task is shown in Table 2.

### 2.3. Survey

During May 2021, a two-part questionnaire collected information on respondents’ background characteristics and DCE preferences. The rate of recruitment of the survey was 95%. A pilot survey was conducted with 20 respondents in order to quality-assure phraseology and question layout. The final version was determined as a two-part questionnaire, of which part one sought respondents’ background characteristics, comprising sex, age (18–30 years old; 31–50 years old; over 50 years old), location (urban or rural based on household registration), educational attainment (low education ≤ 12 years of schooling; medium education between 13 and 16 years of schooling; high education more than 16 years of schooling), job status (farmer; public institutions; company employees; other), self-identified income level (low, medium or high income level) and family relationships (living with the elderly and children). Part two contained 8 binary choice tasks investigating preference trade-offs for vaccine effectiveness, vaccine side effects, duration of protection, vaccination sites, percentage of acquaintances vaccinated and perception of risk. The study was approved by Nanjing Medical University Ethics Committee (No. 2020565).

### 2.4. Sample

The inclusion criteria required all respondents to be over 18 years old and without cognitive impairment. Using a simply random sampling method, 849 respondents in Qingdao were recruited and administered a face-to-face survey by trained researchers, with interviewees given a small monetary token at the completion of the survey. The final sample accounted for a total of 13584 observations, which is sufficient to meet the minimal research sample size requirement set by Orme [70].

### 2.5. Data Analysis

Following commonly used statistical analysis methods for DCE experiments, we adopted the conditional logit model (CLM) to measure individual preferences [41]. In addition, mixed logit models (MLM) and latent class models (LCM) were specified to capture individual preferences heterogeneity. All attribute levels were represented as dummy variables with a selected level for each attribute set as the reference level. Modifying Leng et al.’s model [41], the CLM used to measure individual utility was:
(1)$$\begin{array}{cc} U_{ijs}=\beta_{1}\mathrm{effect}{\left(60\right)}_{ijs} & +\beta_{2}\mathrm{effect}{\left(85\right)}_{ijs}+\beta_{3}\mathrm{sideeffect}{\left(10\right)}_{ijs}\\ & +\beta_{4}\mathrm{sideeffect}{\left(1\right)}_{ijs}+\beta_{5}\mathrm{site}{\left(\sec\mathrm{ondlevel}\right)}_{ijs}\\ & +\beta_{6}\mathrm{site}{\left(\mathrm{thirdlevel}\right)}_{ijs}+\beta_{7}\mathrm{protection}{\left(1\mathrm{yr}\right)}_{ijs}\\ & +\beta_{8}\mathrm{protection}{\left(2\mathrm{yr}\right)}_{ijs}+\beta_{9}\mathrm{acquaintances}{\left(60\right)}_{ijs}\\ & +\beta_{10}\mathrm{acquaintances}{\left(90\right)}_{ijs}+\beta_{11}\mathrm{probinfected}{\left(6\right)}_{ijs}\\ & +\beta_{12}\mathrm{probinfected}{\left(100\right)}_{ijs}+\varepsilon_{ijs} \end{array}$$
where $U_{ijs}$ represents the utility of respondent *i* for scenario *j* in the choice set *s* (where *j* = 1,2; *s* = 1,2,3). *β* is a vector of parameters for each attribute level and utility. $\varepsilon_{ijs}$ is the random utility error.

MLM allow unobserved preference heterogeneity to be modelled through relaxing the independence and irrelevance assumption (IIA) of the error term [71]. We specified that following MLM model:
(2)$$\begin{array}{cc} U_{ijs}=\beta_{1}\mathrm{effect}{\left(60\right)}_{ijs} & +\beta_{2}\mathrm{effect}{\left(85\right)}_{ijs}+\beta_{3}\mathrm{sideeffect}{\left(10\right)}_{ijs}\\ & +\beta_{4}\mathrm{sideeffect}{\left(1\right)}_{ijs}+\beta_{5}\mathrm{site}{\left(\sec\mathrm{ondlevel}\right)}_{ijs}\\ & +\beta_{6}\mathrm{site}{\left(\mathrm{thirdlevel}\right)}_{ijs}+\beta_{7}\mathrm{protection}{\left(1\mathrm{yr}\right)}_{ijs}\\ & +\beta_{8}\mathrm{protection}{\left(2\mathrm{yr}\right)}_{ijs}+\beta_{9}\mathrm{acquaintances}{\left(60\right)}_{ijs}\\ & +\beta_{10}\mathrm{acquaintances}{\left(90\right)}_{ijs}+\beta_{11}\mathrm{probinfected}{\left(6\right)}_{ijs}\\ & +\beta_{12}\mathrm{probinfected}{\left(100\right)}_{ijs}+\varepsilon_{ijs} \end{array}$$
where $U_{ijs}$ represents the utility of respondent *i* for scenario *j* in the choice set *s* (where *j* = 1,2; *s* = 1,2,3).

The LCM is another method for modelling how individual characteristics influence choices [41]. The model assumes that individual preferences are shaped by attributes which may vary across unobservable sub-classes [72]. Following Leng et al. [41], we selected the three-class model with six sociodemographic covariates based on Akaike information criteria (AIC) and Bayesian information criteria (BIC) comparisons across classes and covariates [73]. The central utility function for individual *i* belonging to subclass *q* for scenario *j* in choice sets was modelled through the logit function:
(3)$$\begin{array}{cc} U_{ijs|q}=\beta_{1|q}\mathrm{effect}{\left(60\right)}_{ijs|q} & +\beta_{2|q}\mathrm{effect}{\left(85\right)}_{ijs|q}+\beta_{3|q}\mathrm{sideeffect}{\left(10\right)}_{ijs|q}\\ & +\beta_{4|q}\mathrm{sideeffect}{\left(1\right)}_{ijs|q}+\beta_{5|q}\mathrm{site}{\left(\sec\mathrm{ondlevel}\right)}_{ijs|q}\\ & +\beta_{6|q}\mathrm{site}{\left(\mathrm{thirdlevel}\right)}_{ijs|q}+\beta_{7|q}\mathrm{protection}{\left(1\mathrm{yr}\right)}_{ijs|q}\\ & +\beta_{8|q}\mathrm{protection}{\left(2\mathrm{yr}\right)}_{ijs|q}+\beta_{9|q}\mathrm{acquaintances}{\left(60\right)}_{ijs|q}\\ & +\beta_{10|q}\mathrm{acquaintances}{\left(90\right)}_{ijs|q}+\beta_{11|q}\mathrm{probinfected}{\left(6\right)}_{ijs|q}\\ & +\beta_{12|q}\mathrm{probinfected}{\left(100\right)}_{ijs|q}+\varepsilon_{ijs|q} \end{array}$$

## 3. Results

### 3.1. Sample Characteristics

Table 3 reports key characteristics of the study sample. Out of 849 respondents, the majority of respondents were male (53.4%), aged between 31–50 years old (49%), married (77.7%), on an average income level (75%), from rural areas (64.5%) and had less than 12 years of education (56.7%). As is also shown in Table 3, social relationship dynamic factors were captured, with 12.6% of respondents having older family members at home and 23.56% living with children.

### 3.2. Estimation of Parameters

Table 4 presents the results of the conditional logit model. All variables were statistically significant (*p* < 0.01), except for level 3 vaccination sites and acquaintances vaccinated. Vaccine effectiveness, vaccine-related side effects, followed by duration of protection and perceived probability of infection of individuals/acquaintances were the four most important attributes that influenced individual decision making. Vaccination sites and percentage of acquaintances vaccinated were the two least influential variables. Individuals were more likely to accept the vaccine when it was judged more effective, had fewer side effects and exhibited a longer duration of protection. We also found that respondents reported higher vaccination acceptance rates when there was a high percentage of acquaintances vaccinated around them, and respondents preferred Level 2 township or community health centres over Level 1 village clinics and Level 3 county hospitals and above.

Table 5 presents the results of the MLM. Vaccine effectiveness, vaccine-related side effects, the probability of respondent and acquaintances being infected and the risk of infection were all statistically significant (*p* < 0.01). The percentage of acquaintances vaccinated, when considered as an individual-specific variable, became the third-most dominant force for vaccine acceptance. This suggests that high acquaintance-vaccination rates induce individuals to vaccinate, perhaps appealing to pro-social morals or reflecting peer pressure.

Table 6 presents the results of the LCM. Preference heterogeneity can be clearly seen for different attributes across the three sub-classes. There is strong evidence that supports the existence of latent classes based on socioeconomic covariates, such as age, education, urban/rural residence, income and social relationships. Class 3 was chosen as the reference class and accounted for 31.2% of the total sample. Class 3 also displayed attribute preferences that closely resembled the full conditional logit model.

For Class 2, vaccine side effects, probability of individuals infected and vaccination sites were the three most influential attributes. Duration of protection (more than year) and acquaintances vaccinated (90%) were also statistically significant. From class assignment probabilities, when compared against the reference class, we observe that Class 2 respondents exhibited the characteristics of younger age, lower income, living without elderly family members and coming from rural regions. Our results suggest that Class 2respondents with the above characteristics were less likely to vaccinate.

Class 1respondents were from rural regions and exhibited preferences that were significantly different from the other two classes. Vaccine effectiveness, vaccination sites and proportion of acquaintances vaccinated were all statistically significant. Odd ratios suggest that these attributes significantly influenced preference utility for Class 1individuals. In addition, we observe that the perception of risk also had a profound effect on Class 1 individuals, with vaccination uptake increasing significantly with perception of risk.

Following Leng et al. [41], a marginal analysis was conducted to better understand how vaccination uptake rates change across different attribute levels. We selected a reference vaccine scheme with characteristics set as 40% vaccine effectiveness, 50/100,000 risk of severe side effects, village clinic or community health station administration, protection duration of 6 months, 60% of acquaintances vaccinated and self-determined 100/100,000 risk of infection for individual/acquaintances. Marginal analysis showed that when vaccine effectiveness increased from 40% to 85%, acceptance rates increased by 26%. When vaccine-related side effects reduced from 50/100,000 to 1/100,000, respondents reported a 14.1% increase in willingness to vaccinate. The full marginal analysis results are presented in Figure 1, where the x-axis represents the baseline vaccine scheme.

## 4. Discussion

Compared to the existing vaccination-preference literature conducted under China’s 2021 national vaccination policy [55,56,57,58], our study adopts a broader framework incorporating social relationships (percentage of acquaintances vaccinated and living with the elderly and the young) and cognition (perception of risk). In addition, our study is also one of the first to assess individual vaccination preference changes since the implementation of the 2021 national vaccination policy.

Our results show individuals preferred vaccines that were more effective, had fewer side effects and provided longer protection periods. These results are broadly consistent with the existing pre-2021 and post-2021 national vaccination policy literature [40,41,55,56,57,58]. Similarly to Leng et al.’s pre-national vaccination period study [41], MLM showed that individuals with a higher percentage of acquaintances vaccinated were more likely to vaccinate. ICU workers in the pre-2021 national vaccination policy period also exhibited similar findings [51]. Our LCM revealed clear preference heterogeneity based on social demographic covariates comprising age, education, region, income and social relations (living with the young and old). For Class 1individuals, vaccine effectiveness, vaccination sites and percentage of acquaintances vaccinated significantly influenced vaccination attitudes. This is in contrast to Class 2and three individuals, where β estimates suggest that the marginal effects of these attributes from the mean were less significant. For Class2, younger people with higher education were less likely to vaccinate. This was in line with the results of studies conducted both before and after the implementation of the 2021 national vaccination policy [41,55].

Our study sheds new light on how individual preferences have changed under the 2021 national vaccination policy. In contrast to Leng et al.’s [41] pre-2021 study, the percentage of acquaintances vaccinated was not a statistically significant attribute. A possible explanation is that public concerns regarding the vaccine had diminished through better pro-vaccine education and communication. Further, respondents preferred township community health centres over county hospitals or village clinics. This is in contrast to previous studies, where respondents displayed mistrust towards primary healthcare services and preferred county hospitals [41]. Perhaps community health centres were seen as vaccination sites and county hospitals as disease-treatment centres. In contrast to the pre-2021 study by Liu et al. [55], which found that there was no clear difference in vaccine willingness between rural and urban regions, we found that respondents from Class 1 of our LCM reported that rural regions were more likely to accept the vaccine (*p* < 0.05). We also observed that family urban/rural residency had an effect on Class 2 respondents’ preferences. Importantly, respondents living with elder household members were more likely to vaccinate, whereas those living with the young were not.

We also considered individual cognition as an attribute in our CLM, which has not been assessed in previous studies. Marginal analysis shows that when the perceived probability of infection rose from 1/100,000 to 100/100,000, vaccination uptake probability increased by 11.3%. Vaccine effectiveness, vaccine-related side effects and duration of protection were the three most influential factors alongside the perception of risk. Compared to pre-2021 studies [40,41], where vaccination effectiveness and vaccine-related side effects were the two most important attributes, we found that the importance of protection duration has risen relative to the other factors. This suggests that people’s concerns regarding the vaccine have shifted from safety and effectiveness to longer protection duration. With global research showing the reduction in vaccine effectiveness, particularly against the Omicron variant [74,75], vaccines with longer protection periods were increasingly preferred.

This study has a number of limitations. First, we adopted a simple random sampling method in Qingdao, since COVID-19 travel restrictions restricted inter-city or inter-province face-to-face DCE interviews. As a result, the sample may not have the same representative power as a stratified random sampling method. Second, we controlled for the number of dosages in our study, which reflects the majority of administrated vaccines in China at the time. However, some vaccines administered require more than two dosages for full vaccination. Future preference studies may need to include the number of doses as an attribute and further investigate vaccination preferences for booster doses.

## 5. Conclusions

Our study investigated individual vaccination preferences under China’s 2021 national COVID-19 two-dosage, free and voluntary national vaccination program. The CLM, MLM and LCM showed that in addition to vaccine-specific attributes (vaccine effectiveness, vaccine-related side effects, vaccination sites, duration of protection) and sociodemographic characteristics, social relations (including percentages of acquaintances vaccinated and living with the old) and cognition (perceived risk of infection) were important factors influencing individual preference.

Our study compared findings with previous research conducted pre-2021 national vaccination policy. We found that under the 2021 national vaccination policy, individuals continued to prefer vaccines with better efficacy, longer protection periods and higher safety standards. We also discovered that when compared to studies conducted in the pre-2021 period, safety concerns about the vaccine have diminished and respondents’ preferences have shifted towards longer protection and accessibility.

Our study will contribute to the current efforts in the vaccine rollout and inform the government in developing more effective vaccination polices in the future. With many Chinese provinces starting to implement booster doses, we believe that vaccination policies aiming to increase accessibility, targeting individuals with higher risk of infection and emphasising individual risk will result in higher vaccination uptake. Health authorities should be aware that a high acquaintance-vaccination rate induced individuals to vaccinate. Social relationships, especially the desire to protect elders from infection when living together, should be a feature of information campaigns in the rollout of China’s vaccine programs.

## Figures and Tables

**Figure 1 vaccines-10-00543-f001:**
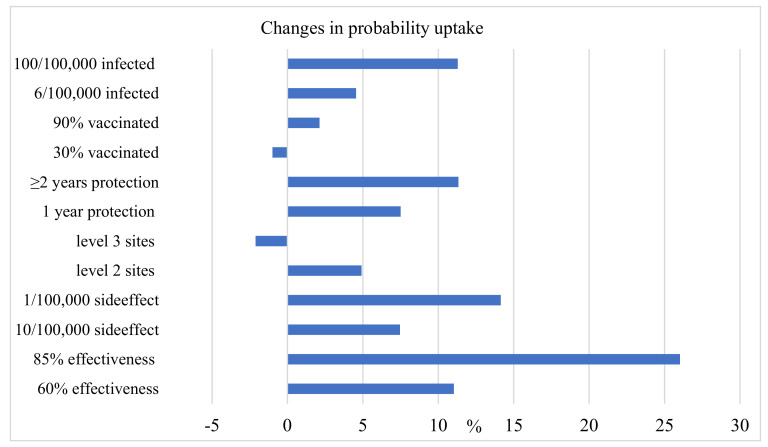
Changes in probability uptake.

**Table 1 vaccines-10-00543-t001:** Attributes and Levels Used in the Discrete Choice Experiment.

Attributes	Levels	Descriptions
Vaccine effectiveness	40%	Protects 40% of vaccinated
	60%	Protects 60% of vaccinated
	85%	Protects 85% of vaccinated
Self-assessed vaccine-related side effects	50/100,000	50 out of 100,000 risk of severe side effects
	10/100,000	10 out of 100,000 risk of severe side effects
	1/100,000	1 out of 100,000 risk of severe side effects
Vaccination sites	Level 1	village clinic or community health station
	Level 2	township or community health centre
	Level 3	county hospital and above
Duration of vaccine protection	six months	Six months of vaccine protection
	one year	One year of vaccine protection
	More than two years	More than two years
Acquaintances vaccinated	30%	30% of your family, friends and acquaintances already vaccinated
	60%	60% of your family, friends and acquaintances already vaccinated
	90%	90% of your family, friends and acquaintances already vaccinated
Risk perception (probability including yourself and acquaintances being infected with COVID-19)	100/100,000	100 out of 100,000 contracting COVID-19
	6/100,000	6 out of 100,000 contracting COVID-19
	1/100,000	1 out of 100,000 contracting COVID-19

**Table 2 vaccines-10-00543-t002:** Choice set.

Q1	Vaccine A	Vaccine B
Vaccine effectiveness	40%	60%
Vaccine-related side effects	10/100,000	50/100,000
Vaccination sites	Level 1	Level 2
Duration of vaccine protection	one year	six months
Acquaintances vaccinated	30%	60%
The probability of infection with COVID-19	100/100,000	6/100,000
Which vaccine do you prefer?		

**Table 3 vaccines-10-00543-t003:** Characteristics of the study sample (*n* = 849).

Characteristics	*n*	%
Sex		
Male	453	53.357
Female	396	46.643
Age		
Age 18–30	168	19.788
Age 31–50	416	48.999
Age 51+	265	31.213
Marital status		
married	660	77.739
unmarried/widowed/divorced	189	22.261
Residence		
urban area	301	35.453
rural area	548	64.547
Years of education		
low education (≤12 years)	481	56.655
medium education (13— ≤16years)	263	30.978
high education (>16 years)	105	12.367
Occupation		
farmer	257	30.271
government/public institution staff	180	21.201
company employees (including migrant workers, individual businesses, etc.)	290	34.158
Other (including retired, student)	122	14.370
Household yearly income		
low income level	114	13.428
medium income level	637	75.029
high income level	98	11.543
Elderly at home	107	12.603
Children at home	200	23.557

**Table 4 vaccines-10-00543-t004:** Conditional logit model of respondent preferences.

Attribute	ß	SE	*p* Values	95% CI
Vaccine effectiveness (reference = 40%)				
60%	0.423	0.036	0.000	0.351, 0.494
85%	0.806	0.041	0.000	0.727, 0.886
Vaccine-related side effects (reference = 50/100,000)				
10/100,000	0.251	0.035	0.000	0.182, 0.320
1/100,000	0.432	0.037	0.000	0.358, 0.507
Vaccination sites (reference = Level 1)				
level 2	0.141	0.037	0.000	0.067, 0.214
level 3	−0.067	0.036	0.063	−0.138, 0.004
Duration of vaccine protection (reference = six months)				
one year	0.245	0.037	0.000	0.173, 0.316
more than two years	0.350	0.036	0.000	0.279, 0.421
Acquaintances vaccinated (reference = 30%)				
60%	0.031	0.037	0.409	−0.042, 0.103
90%	0.093	0.037	0.011	0.021, 0.165
The probability of respondents/acquaintances infected (reference = 1/100,000)				
6/100,000	0.221	0.037	0.000	0.148, 0.293
100/100,000	0.346	0.036	0.000	0.274, 0.417
Model fit				
Observations = 13,584				
Respondents = 849				
Prob > chi2 = 0.000				
Pseudo R2 = 0.0857				
LR chi2(13) = 4304.62				
AIC = 8633.24				
BIC = 8723.44				

**Table 5 vaccines-10-00543-t005:** Mixed Logistic Regression Models of Patient Preferences.

Variables	ß	SD	*p* Values	95% CI
Vaccine effectiveness (reference = 40%)				
60%	0.543	0.005	0.001	0.465, 0.621
85%	1.537	1.164	0.001	1.418, 1.656
Vaccine-related side effects (reference = 50/100,000)				
10/100,000	0.658	0.535	0.001	0.581, 0.736
1/100,000	1.402	1.326	0.001	1.285, 1.519
Vaccination sites (reference = level 1)				
level 2	0.055	0.885	0.187	−0.027, 0.137
level 3	−0.361	0.637	0.001	−0.445, −0.276
one year	−0.048	−0.268	0.197	−0.122, 0.025
more than two years	0.096	0.408	0.013	0.020, 0.171
Acquaintances vaccinated (reference = 30%)				
60%	0.226	0.014	0.001	0.149, 0.304
90%	0.440	0.058	0.001	0.361, 0.519
6/100,000	0.252	−0.392	0.001	0.175, 0.329
100/100,000	0.374	0.744	0.001	0.290, 0.459

**Table 6 vaccines-10-00543-t006:** Latent class logit model of patient preferences.

Attribute	Class 1			Class 2			Class 3		
	ß	SE	*p* Value	ß	SE	*p* Value	ß	SE	*p* Value
Vaccine effectiveness (reference = 40%)									
60%	3.007	0.333	0.000	−0.052	0.080	0.502	0.902	0.114	0.000
85%	6.344	0.771	0.000	0.060	0.086	0.488	1.351	0.149	0.000
Vaccine-related side effects (reference = 50/100,000)									
10/100,000	−0.027	0.390	0.946	0.633	0.070	0.000	0.261	0.089	0.000
1/100,000	0.093	0.349	0.791	1.247	0.095	0.000	−0.442	0.119	0.000
Vaccination sites (reference = first level)									
second level	3.317	0.859	0.000	0.276	0.072	0.000	−0.388	0.112	0.000
third level	−2.244	0.278	0.000	−0.181	0.079	0.021	0.684	0.113	0.000
Duration of vaccine protection (reference = six months)									
one year	1.732	0.264	0.000	0.082	0.073	0.259	0.098	0.105	0.350
more than two years	0.475	0.326	0.145	0.197	0.072	0.005	0.991	0.116	0.000
Acquaintances vaccinated (reference = 30%)									
60%	−1.259	0.366	0.001	0.081	0.067	0.228	0.044	0.103	0.679
90%	−2.354	0.348	0.000	0.474	0.075	0.000	−0.112	0.107	0.293
The probability of individuals/acquaintances infected (reference = 1/100,000)									
6/100,000	0.667	0.476	0.117	−0.536	0.077	0.000	0.10	0.111	0.363
100/100,000	2.588	0.554	0.000	−1.133	0.088	0.000	−0.081	0.093	0.386
Class probability model									
age	0.041	0.159	0.795	−0.281	0.131	0.032	-	-	-
education	0.213	0.181	0.239	0.128	0.144	0.374	-	-	-
urban/rural residence	0.477	0.238	0.045	0.528	0.186	0.005	-	-	-
average monthly household income	−0.185	0.209	0.374	−0.431	0.170	0.011	-	-	-
elderly at home	−0.131	0.297	0.658	−0.746	0.316	0.018	-	-	-
children at home	−0.110	0.266	0.678	−0.030	0.232	0.870	-	-	-
constant	−1.126	0.772	0.144	0.894	0.525	0.089	-	-	-
Class probability									
Average	0.220			0.468			0.312		
Model fit									
Observations = 13584									
Respondents = 849									
AIC = 7740.432									
BIC = 7977.635									

## Data Availability

The data presented in this study are available on request from the corresponding author.

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
