# Peer review of "Individual Preferences for COVID-19 Vaccination under the China’s 2021 National Vaccination Policy: A Discrete Choice Experiment Study"

_vaccines, 2022, doi:10.3390/vaccines10040543_

Round 1

Reviewer 1 Report

Overall this is an interesting study.  Public health groups may have difficulty understanding all of the statistical analyses and especially the discussion related to Class 1, 2 and 3 since these are not easily teased out to determine who they describe.

There seem to be some missing lines from tables.

The analyses are complex but usually the discussion of the tables explains most of the results.

Specific comments:

Line 100--how confident are the authors that the participants understand numerical presentations and percent---e.g. 1/100000 vs 6/100000 vs 100/10000 or 30% vs 45%, etc.  Are the authors assuming it is just the relative size of numerators or numbers of percents that are important for comparisons?  Even among highly educated people, numeracy can be low.

Line 112-Table 1.  The "severe" shows up here but is not defined.  Was this defined for the participants or was it left to them to define?

Line 112 Table 1---do all of the site pertain to all participants?  For urban participants, does community site make sense?  Did you stratify choices by residence of the participant?

Line 124--did pilot study assess ability to understand fractions and percents?

Line 139---was was the rate of recruitment?  Did 100% of those invited complete the survey?

Line 199-Table 4---line for probability of acquaintances infected require a reference level--1/100000?

Line 206--Table 5---are some reference levels missing here?

Line 251---the discussion of the importance of Class is difficult for the less statistically savvy reader to understand.  What are the characteristics of Class 1 vs 2 vs 3 individuals?

Line 256---could the move away from desiring vaccine at county hospital have to do with this being the site where severe infections are treated and where many COVID deaths occur?

Line 274---is "preceived risk" really "cognitive level"?  You state that only those without cognitive issues were invited.

Author Response

Response to Reviewer 1 comments:

1. Line 100--how confident are the authors that the participants understand numerical presentations and percent---e.g. 1/100000 vs 6/100000 vs 100/10000 or 30% vs 45%, etc.  Are the authors assuming it is just the relative size of numerators or numbers of percents that are important for comparisons?  Even among highly educated people, numeracy can be low.

Response: Trained interviewers who administered the survey were able to provide advice.

2. Line 112-Table 1.  The "severe" shows up here but is not defined.  Was this defined for the participants or was it left to them to define?

Response: Severe was self-defined by the participants, which is noted in Table 1.

3. Line 112 Table 1---do all of the site pertain to all participants?  For urban participants, does community site make sense?  Did you stratify choices by residence of the participant?

Response: All of the sites pertain to all participants. Community health centres are common in both urban and rural regions in China. We did not stratify choices by residency, but we did collect residency information to analysis possible preference heterogeneity.

4. Line 124--did pilot study assess ability to understand fractions and percents?

Response: Yes. No problem found with the fractions/percentages. Trained interviewers who administered the survey were able to provide advice.

5. Line 139---was was the rate of recruitment?  Did 100% of those invited complete the survey?

Response: We did not formally collect the rate of recruitment. A conservative estimate of 95% rate of recruitment was added in line 125.

6. Line 199-Table 4---line for probability of acquaintances infected require a reference level--1/100000?

Response: added reference level to line 201-Table 4

7. Line 206--Table 5---are some reference levels missing here?

Response: Vaccine effectiveness, vaccine side-effects, duration of vaccine protection, acquaintances vaccinated and probability of individuals/acquaintances infected have reference levels. All attributes have descriptions.

8. Line 251---the discussion of the importance of Class is difficult for the less statistically savvy reader to understand.  What are the characteristics of Class 1 vs 2 vs 3 individuals?

Response: Class 3 was used as the reference class as noted in line 212. We have added statistical explanations for class 2 respondents’ characteristics in lines 219-223 and explanations for class 1 respondents’ characteristics in line 224.

9. Line 256---could the move away from desiring vaccine at county hospital have to do with this being the site where severe infections are treated and where many COVID deaths occur?

Response: We have added on lines 271-272: “Perhaps community health centers were seen as vaccination sites and county hospitals as disease treatment centers.”

10. Line 274---is "perceived risk" really "cognitive level"?  You state that only those without cognitive issues were invited.

Response: Supported by the references in line 58, we used “cognition” as an individual’s ability to understand, process information and assess the factors and risks of Covid infection and vaccine attributes. Please also see the definition on line 46.

Reviewer 2 Report

This is an elegant study that employs sophisticated modelling on a dataset of adequate size to reliably enable a determination, for a post-2021 cohort, of factors that significantly influenced vaccine uptake. The study, one of the first to assess individual vaccination preference changes since the implementation of the 2021 national vaccination policy in China shows that individuals preferred vaccines that were more effective, had fewer side-effects and provided longer periods of protection. It is interesting that in contrast to the findings in the pre-2021 study of Leng et al (2021) public concerns for the vaccine had diminished which may be related to improved pro-vaccination education and communication.

The survey instrument design and choice of analytical tools and modelling seems to be very well informed. Methods and results presented in Tables 1 to 6 clearly enable the reader to appreciate the factors of significance. In the MLM results presented in Table 5 vaccine effectiveness, vaccine related side effects and probability of respondents and acquaintances being infected and risk of infection were all identified as statistically significant while the results for the LCM in Table 6 revealed strong evidence for the existence of latent classes based upon socioeconomic covariates.

Once again, differences between some findings in this study and those reported for studies on pre-2021 cohorts, eg the paper of Liu et al (2021), do seem to be consistent with improved vaccine education and communication which should certainly be an important guide to continuing vaccination strategies as the COVID experience continues!

The concluding paragraph of the paper does contain wise recommendations that can be made based upon the findings in this paper and those in several recent papers that have reported on vaccine acceptance, hesitancy and the role of sound community education initiatives.

The authors helpfully report a number of limitations for this study that can guide informed design of future studies. For some vaccines a fourth dose is often now recommended which will introduce yet another factor to be considered in studies seeking to investigate vaccine acceptance.

Minor correction

L227  ….how vaccination uptake rates change across …

L262 not a statistically significant attribute.

Fig 1. Legend:  Changes in probability uptake

Author Response

Response to Reviewer 2 comments:

1. L227….how vaccination uptake rates change across …

Response: corrected grammatical error in line 231 of the updated manuscript.

2. L262 not a statistically significant attribute.

Response: Corrected grammatical error in line 266 of the updated manuscript.

3. Fig 1. Legend:  Changes in probability uptake.

Response: Corrected grammatical error in Figure 1 legend.